# Clinical course and risk factors for sleep disturbance in patients with ischemic stroke

Hui-Ju Tsai[1], Yi-Sin Wong[2], Cheung-Ter Ong[3]*

1 Department of Internal Medicine, Ditmanson Medical Foundation Chia-Yi Christian Hospital, Chia-Yi, Taiwan, 2 Department of Family Medicine, Ditmanson Medical Foundation Chia-Yi Christian Hospital, Chia-Yi, Taiwan, 3 Department of Neurology, Ditmanson Medical Foundation Chia-Yi Christian Hospital, Chia-Yi, Taiwan

* ctong98@yahoo.com.tw

## Abstract

### Background

Studies on insomnia in patients with ischemic stroke, particularly in the acute phase, are limited. The proportion of patients with sleep disturbance during the acute stroke period who are likely to develop insomnia in subacute and chronic stages of stroke is unknown. This study aimed to investigate the risk factors for sleep disturbance and the clinical course of the disease in patients with acute ischemic stroke.

### Methods

This prospective observational study included patients diagnosed with ischemic stroke between July 1, 2020, and October 31, 2021. The Diagnostic and Statistical Manual of Mental Disorders, Fifth Edition (DSM-5) for insomnia and the eight-item Athens Insomnia Scale (CAIS-8) were used to diagnose insomnia. Beck Depression Inventory (BDI) was applied to evaluate the mood of patients. Patient reported their sleeping conditions, before stroke onset and during the acute (within 7 days) and chronic (3 months after presentation) stroke periods.

### Results

In total, 195 patients with ischemic stroke were included in this study. Of these, 34.3% (67), 37.4% (73), and 29.7% (58) presented with sleep disturbance before stroke onset and during the acute and chronic stroke periods, respectively. Of the 128 patients without insomnia before stroke onset, 15.6% (20/128) presented with insomnia symptoms 3 months after stroke onset. Moreover, 13 (12.7%) of the 102 patients without sleep disturbance during the acute stroke period developed insomnia 3 months after stroke onset. Of the 67 patients with insomnia before stroke onset 29 (43.3%) did not develop the condition 3 months after stroke onset. A higher risk of sleep disturbance was associated with atrial fibrillation, hypertension, and mood disturbance in the acute stroke period, and a higher risk of insomnia was associated with low education and mood disturbance in the chronic stroke period.

**Data Availability Statement:** All relevant data are within the paper.

**Funding:** This research was supported by Ditmanson Medical Foundation Chia-Yi Christian Hospital Research grant (R109-14). The funders

had no role in study design, data collection and analysis, decision to publish, or preparation of the manuscript.

**Competing interests:** The authors have declared that no competing interest exist.

## Conclusion

The prevalence rates of sleep disturbance before and during the acute and chronic stroke periods were 34.3%, 37.4%, and 29.7%, respectively. The incidence of stroke-related insomnia was 15.6%. Patients with insomnia before stroke may recover after the stroke. Atrial fibrillation, hypertension, and mood disturbance were associated with a higher risk of sleep disturbance in the acute stroke period, whereas low education and mood disturbance were associated with insomnia in the chronic stroke period.

## Introduction

The incidence of stroke and the rate of stroke-related mortality have decreased over the last two decades. However, stroke remains a leading cause of mortality and morbidity [1, 2]. Owing to changes in demographic characteristics, including population aging and health transition in developing countries, the absolute number of patients who develop stroke annually and that of survivors with sequelae is expected to increase in the future [3]. Stroke survivors often experience long-term symptoms correlated with stroke sequelae, such as motor function, cognition, and language and communication impairment, mood and activity of daily living disturbance, and social isolation [4–6]. Insomnia is a common symptom among patients with ischemic stroke. The prevalence of insomnia among patients with stroke is 20%–50% [7]. Leppävuori et al. showed that 37.5% of patients with stroke developed insomnia 3 months after onset and that 18.1% of insomnia cases were correlated with stroke [8]. According to the Diagnostic and Statistical Manual of Mental Disorders, Fifth Edition (DSM-5), insomnia is defined as dissatisfaction with sleep quality or quantity and presents with at least one of the following symptoms: difficulties in initiating sleep, challenges in maintaining sleep, and early-morning awakening with inability to return to sleep [9]. In addition to these symptoms, sleeping difficulties should last >3 months and occur at least 3 nights per week. The prevalence rate of insomnia in the general population varies from 6% to 40% [10–12]. When patients presented with symptoms of sleep disorder without fulfilling all insomnia criteria of DSM-5, they were considered to exhibit insomnia symptoms [13].

Sleep disorders are a common neurological complication in the initial stage of acute stroke and include hypersomnia, daytime sleepiness, and insomnia [14]. Moreover, insomnia is associated with a higher risk of ischemic stroke [15]. Stroke-related insomnia commonly occurs in the early stage of stroke [16]. The development of insomnia may be correlated with stroke location, emotional stress, and environmental effect. During the acute stroke period, two dysfunctional areas affect neurological function, which may affect sleep quality in patients with ischemic stroke. One is the core lesion, where the lack of blood flow results in irreversible neuronal death. The other is the penumbra, where the cell is viable but the blood circulation is insufficient [17, 18]. After stroke onset, functional impairment occurs not only because of the presence of lesions in the central core and penumbra but also because of disturbance in the interplay between the ischemic core and adjacent and remote areas. Therefore, disturbance in this connection may affect sleep quality or pattern. Whether the neurological impairment in the acute phase of stroke may exacerbates the risk of insomnia and whether insomnia during the acute phase stroke period improves after reperfusion in the penumbra requires further investigation.

A previous study has shown that compared with the controls, patients with stroke had a longer sleep latency and poorer sleep quality. Moreover, patients with stroke had a greater awake time in the night and wake efficiency than controls [19]. He et al. established that insomnia in the acute stroke period may impair the diastolic function of cerebral arteries and damage the cerebrovascular reserve function. These mechanisms can subsequently affect the recovery of neurological function and increase the risk of stroke recurrence [20]. Insomnia has a negative impact on the outcome of patients with stroke. A previous study has proven that patients without insomnia exhibit a lower depression score and greater power and independence than those with stroke who developed insomnia [21]. Insomnia in patients with stroke may affect recovery and is associated with a higher risk of stroke recurrence [22]. Studies on insomnia among patients with ischemic stroke, particularly in the acute phase are limited, compared with those on the general population. Most studies have investigated the prevalence of insomnia in patients with ischemic conditions and have assessed sleep quality after the onset of stroke. In patients with ischemic stroke, prospective data on sleep quality before stroke (acute stroke and subacute stroke) are not available. Hence, this study aimed to investigate the prevalence of insomnia and factors affecting sleep quality among patients with acute-phase ischemic stroke. The study further intended to evaluate the proportion of patients who developed insomnia during the acute and chronic stroke periods [23].

## Methods

### Participants

This was a prospective observational study. Patients admitted to the neurology department of a teaching hospital in Central Taiwan were recruited sequentially. Insomnia in patients with first ischemic stroke was evaluated using a questionnaire about insomnia and depression.

### Sampling

Consecutive patients without a stroke history who were diagnosed with ischemic stroke and admitted to the hospital between July 1, 2020, and October 31, 2021, were included in the analysis. Clinical neurological examination and brain computed tomography scan were performed at the emergency department. Magnetic resonance imaging (MRI) was performed on the second or third day of stroke onset. In the clinical examination, the National Institute of Health Stroke Scale, Mini-Mental State Examination (MMSE), and modified Ranking Score (mRS) were used. Stroke subtypes were classified in accordance with the TOAST criteria [24]. The stroke locations were categorized into the right anterior circulation, left anterior circulation, right posterior circulation, left posterior circulation, and multiple (involving more than one circulation territory) [8]. The DSM-5 criteria were used to diagnose insomnia [25]. All participants provided written informed consent. Patients aged between 20 and 80 years, those with acute-onset ischemic stroke confirmed by a neurologist, those with first stroke event, those who arrived at the hospital within 48 h after stroke onset, and those with stroke confirmed on brain MRI were included. Patients who could not communicate and those with severe dementia (MMSE score of <10), congestive heart failure, chronic obstructive pulmonary disease or asthma, and major depression and regular antidepressant therapy as well as those who did not provide informed consent were excluded.

Participants were interviewed at the stroke unit of the hospital. Written informed consent was obtained from all patients. The interviewer received comprehensive training, including that on communication with elderly individuals.

The baseline demographic, clinical, and medical characteristics of participants were recorded by a study nurse. All participants were interviewed face to face by a trained

interviewer, and they reported their Chinese of the eight-item Athens Insomnia Scale (CAICS-8) and Beck Depression Inventory (BDI) during stroke onset and the acute (within 7 days after onset) and chronic (3 months after onset) stroke periods.

## Insomnia

Insomnia was confirmed using CAIS-8. The Cronbach's α of internal consistency between the Chinese version of the eight-item Athens Insomnia Scale (AIS) and Insomnia Self-assessment Inventory was 0.82 [26]. The AIS had a good sensitivity and specificity for insomnia and exhibit significant correlations with other insomnia scales [27]. The Insomnia Self-assessment Inventory has eight items, namely, sleep induction, awakenings in the night, final awakening earlier than desired, total sleep duration, overall quality of sleep, sense of well-being during the day, functioning during the day, and sleepiness during the day. Each item was rated from 0 to 4 (0 = no problem or normal, 3 = very severe problem).

We evaluated the sleep condition before and during the acute and chronic stage of stroke. During acute hospitalization, patients reported their CAIS-8 for sleep condition before stroke onset and during the acute and chronic stroke periods.

The total score was obtained by summing up all responses, which ranged from 0 to 24. The cutoff score for insomnia was 8. In this study, a score of $\geq 8$ indicated insomnia.

In addition to a CAIS-8 of $\geq 8$, the diagnosis of insomnia before and 3 months after stroke onset was based on at least one of the following symptoms: difficulties in initiating sleep, challenges in maintaining sleep, early-morning awakening with inability to return to sleep, and sleeping difficulties lasting for more than 3 months and occurring at least 3 nights per week. The diagnosis of sleep disturbance in the acute stroke period was based on the same criteria as those for insomnia before and 3 months after stroke onset. Insomnia was considered when patients presented with symptoms of difficulties in initiating sleep, challenges in maintaining sleep, or early-morning awakening with inability to return to sleep during acute stroke period.

## Mood evaluation

BDI is a 4-point scale with 21 self-reported items. The scale has a high reliability and has been widely used for evaluating patients with depression [28, 29]. In this study, patients with stroke reported their BDI score within the first week of stroke and 3 months after stroke onset. A BDI score of $> 10$ indicated mood disturbance [29, 30]. This study was approved by the Institutional Review Board (IRB) of the hospital (CYCH-IRB: 2020026).

## Statistical analysis

The proportion of patients presenting with each factor and the rate of patients with sleep disorders were expressed as percentages. Age was presented as mean and standard deviation. The chi-square test or the Fisher's exact test was used to analyze categorical variables (stroke type, location, and risk factors). Binary logistic regression analysis was performed to assess the risk factors of sleep disturbance in acute and chronic stroke period. Sex, atrial fibrillation, hypertension, diabetes mellitus, chronic kidney disease, living with spouse, education level, smoking habit, alcohol consumption, benut (areca catechu) habit, and mood disturbance were used as predictors. Odds ratios with 95% confidence intervals were obtained from binary logistic regression, which were used to show the correlation between stroke risk factors and sleep disorders. All statistical analyses were performed using a commercially available software (Statistical Package for the Social Sciences software version 21.0., IBM Corporation, Somers, NY, USA). Two-sided P values of $< 0.05$ were considered statistically significant.

## Results

In total, 595 patients diagnosed with ischemic stroke were admitted to our hospital between July 1, 2020, and October 31, 2021. However, the following patients were excluded from the study: those who died within 3 months after stroke onset (n = 18), those who were hospitalized for < 7 days (n = 31), those without evidence of stroke on brain MRI (n = 50), those with stroke recurrence (n = 88), those who did not agree to participate (n = 41), those with dementia (an MMSE score of < 10; n = 7), those with other medical conditions (such as sepsis, encephalitis and cancer, n = 45), those with depression despite regular pharmacological treatment (n = 35), those who could not clearly express their opinions (n = 36), and those aged > 80 years (n = 49). Finally, 195 patients were included in the analysis. Table 1 lists the characteristics of patients. Of the 195 patients with ischemic stroke, 67 (34.4%) presented with insomnia before stroke onset, whereas 128 (65.6%) did not. Of the 67 patients with insomnia before stroke onset, 47 (70.1%) developed the symptoms during the acute stroke period and 20 (29.9%) patients did not. In total, 26 (20.3%) of the 128 patients without insomnia before stroke onset developed the symptoms during the acute stroke period, and 102 (79.7%) did not. At 3 months after stroke onset, 58 patients had insomnia, and 147 did not. Moreover, 56.7% (38/67) of 67 patients with insomnia before stroke onset presented with the problem 3 months after stroke onset. However, 29 (43.3%) patients did not. Of the 128 patients without insomnia before stroke onset, 20.3% (26/128) presented with insomnia symptoms during the acute stroke period. Totally, 26.9% (7/26) of the patients experienced insomnia 3 months after stroke

**Table 1. Demographic characteristics of patients and manifestations of stroke (n = 195).**

| | |
|---|---|
| Age, mean ± SD (years) | 64.1 ±8.9 |
| Sex (F), % | 40.5% (n = 79) |
| Education | |
| 0–6 years | 45.7% (n = 89) |
| 7–12 years | 41.5% (n = 81) |
| > 12 years | 12.8% (n = 25) |
| Married, % | 95.9% (n = 187) |
| Living with spouse | 74.8% (n = 142) |
| Not living with spouse | 25.2% (n = 53) |
| Smoking, % | 25.1% (n = 49) |
| Alcohol use, % | 20.1% (n = 40) |
| Benut use, % | 9.7% (n = 19) |
| Hypertension, % | 80.0% (n = 156) |
| Diabetes mellitus, % | 46.7% (n = 91) |
| Atrial fibrillation, % | 6.7% (n = 13) |
| Use of sleep-promoting agent | 10.8% (n = 21) |
| Stroke localization | |
| Right anterior circulation, % | 35.4% (n = 69) |
| Left anterior circulation, % | 33.3% (n = 65) |
| Right posterior circulation, % | 18.0% (n = 35) |
| Left posterior circulation, % | 13.3% (n = 26) |
| Stroke severity | |
| NIHSS score of 0–6 | 80.5% (n = 157) |
| NIHSS score of 7–13 | 19.0% (n = 37) |
| NIHSS score of > 13 | 0.5% (n = 1) |

Benut: Areca catechu, NIHSS: National Institute Health Stroke Scale.

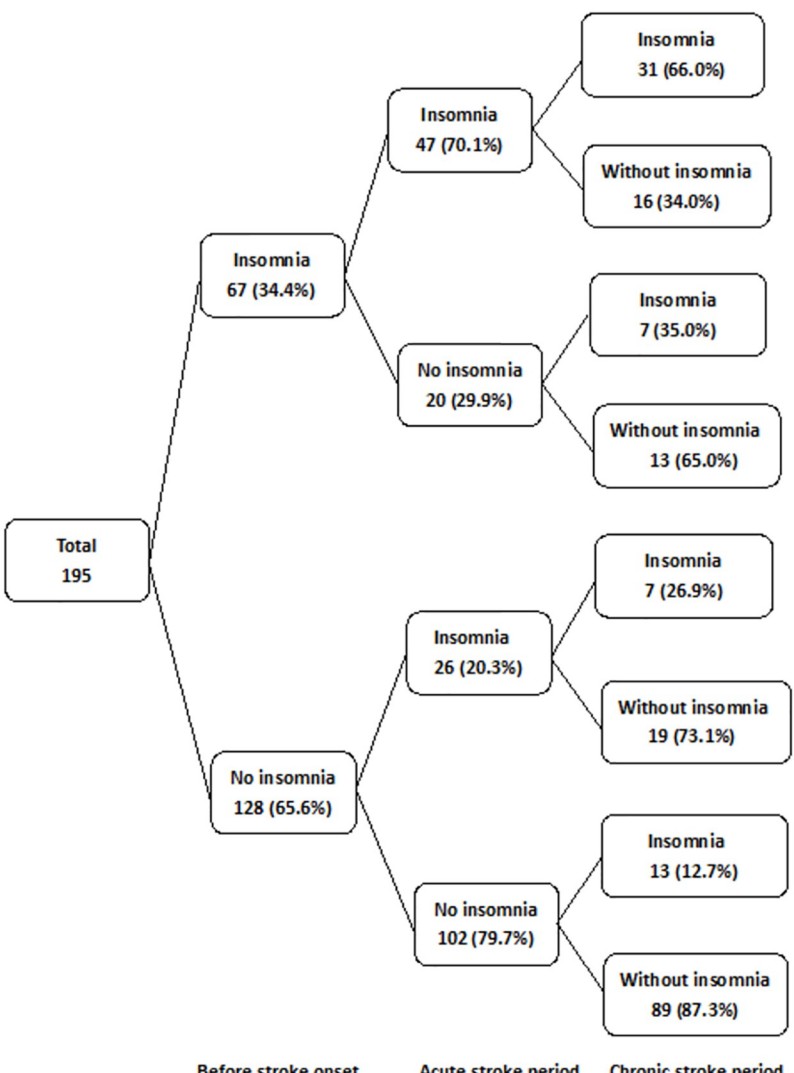

**Fig 1. Sleep conditions in patients with ischemic stroke Insomnia in acute stroke period: Insomnia symptoms.**

onset. Of the 122 patients without insomnia symptoms during the acute stroke period, 16.4% (20/122) presented with insomnia 3 months after stroke onset. Furthermore, 20 patients had insomnia 3 months after stroke onset and 108 did not. The prevalence rates of sleep disturbance were 34.3% (n = 67), 37.4% (n = 73), and 29.7% (n = 58) before stroke onset and during the acute and chronic stroke periods, respectively (Fig 1). In total, 20 (15.6%) of the 128 patients did not present with insomnia before stroke onset and developed the problem at 3 months after stroke onset. Hence the insomnia was considered to be stroke-related. Stroke was not always correlated with a higher risk of insomnia. Approximately 29 (43.3%) of the 67 patients who had insomnia before stroke onset did not present with the condition 3 months after stroke onset.

## Factors affecting sleep disorders

Univariate analysis indicated that the risk factors for stroke including diabetes mellitus, hypertension, dyslipidemia, and atrial fibrillation, did not affect the development of insomnia in

**Table 2. The relationship between sleep disorder and type of stroke (n = 195).**

| | Acute stroke period | | | Chronic stroke period | | |
|---|---|---|---|---|---|---|
| | Sleep disorder | No sleep disorder | p | Sleep disorder | No sleep disorder | p |
| Stroke type | | | | | | |
| SAO | 34 (32.7%) | 70(68.3%) | 0.46 | 33(31.7%) | 71(68.3%) | 0.56 |
| LAA | 12(36.4%) | 21(63.6%) | | 8(24.2%) | 25(75.8%) | |
| CAE | 5(55.6%) | 4(44.4%) | | 3(33.3%) | 6(66.7%) | |
| SOU | 21(45.7%) | 25(54.3%) | | 12(26.1%) | 34(73.9%) | |
| SOD | 1(33.3%) | 2(66.7%) | | 2(66.7%) | 1(33.3%) | |
| Location of stroke | | | | | | |
| RAC | 25(36.2%) | 44(63.8%) | 0.96 | 14(20.3%) | 55(79.7%) | 0.06 |
| LAC | 25(38.5%) | 40(61.5%) | | 20(30.8%) | 45(69.2%) | |
| PCC | 23(37.7%) | 38(62.3%) | | 24(39.3%) | 37(60.7%) | |

SAO: small-artery occlusion, LAA: large-artery atherosclerosis, CAE: cardioembolism, SOU: stroke of undetermined etiology, SOD: stroke of other determined etiology.
RAC: right anterior cerebral circulation, LAC: left anterior cerebral circulation, PCC: posterior cerebral circulation.

patients during the acute and chronic stroke periods. Moreover, stroke severity, subtype and location did not affect the risk of insomnia (Table 2). Mood disturbance was associated with a higher risk of insomnia symptoms in the acute stroke period and mood disturbance and school education for $\leq 6$ years were linked to risk of insomnia in the chronic stroke period (Table 3). Multivariate analysis showed that atrial fibrillation, hypertension, benut habit and mood disturbance were correlated with a higher risk of insomnia symptoms during the acute insomnia stage. Furthermore, low education and mood disturbance were risk factors for insomnia 3 months after stroke onset (Table 4).

## Discussion

This study resulted in the following findings: Approximately 16% of patients developed insomnia after stroke onset. Insomnia may improve after stroke onset among patients presenting

**Table 3. Factors affecting insomnia in patients with ischemic stroke (n = 195).**

| | Insomnia symptom during the acute stroke stage | | | Insomnia at 3 months | | |
|---|---|---|---|---|---|---|
| | No (n = 135) | Yes(n = 60) | P value | No (n = 137) | Yes(n = 58) | p value |
| Sex (M) | 80(59.3%) | 36(60%) | 0.50 | 86(62.8%) | 30(51.7%) | 0.10 |
| Atrial fibrillation | 7(5.2%) | 6(10%) | 0.17 | 6(4.4%) | 7(12.1%) | 0.05 |
| Hypertension | 106(78.5%) | 50(83.3%) | 0.28 | 110(80.3%) | 46(79.3%) | 0.51 |
| Diabetes mellitus | 65(48.1%) | 26(43.3%) | 0.32 | 63(46.0%) | 28(48.3%) | 0.45 |
| Chronic kidney disease | 14(10.4%) | 7(11.7%) | 0.48 | 18(13.1%) | 3(5.2%) | 0.07 |
| Living with spouse | 96(71.1%) | 46(76.7%) | 0.27 | 104(75.9%) | 38(65.5%) | 0.09 |
| Education of > 6 years | 75(55.6%) | 32(53.3%) | 0.45 | 85(62.0%) | 22(37.9%) | 0.002[#] |
| Smoking | 31(23.0%) | 18(30%) | 0.19 | 34(24.8%) | 15(25.9%) | 0.51 |
| Alcohol | 28(20.7%) | 12(20%) | 0.54 | 29(21.2%) | 11(19.0%) | 0.45 |
| Benut | 12(8.9%) | 7(11.7%) | 0.36 | 10(7.3%) | 9(15.5%) | 0.07 |
| Mood disturbance | 7(5.2%) | 9(15%) | 0.03[#] | 9(6.6%) | 14(24.1%) | 0.001[#] |

Benut: Areca catechu,
[#]: p < 0.05.

**Table 4. Risk factors for insomnia among patients with ischemic stroke (binary logistic regression).**

| | Insomnia during the acute stroke stage | | Insomnia at 3 months | |
|---|---|---|---|---|
| | P | OR (95% CI) | P | OR (95% CI) |
| Sex(female) | 0.29 | 1.53 (0.69–3.40) | 0.51 | 1.33 (0.57–3.10) |
| Atrial fibrillation | 0.002 | 8.64 (21.8–34.2)[#] | 0.24 | 2.23(0.58–8.48) |
| Hypertension | 0.03 | 2.69(1.03–6.42)[#] | 0.86 | 0.93 (0.39–2.16) |
| Diabetes mellitus | 0.53 | 0.81(0.43–1.55) | 0.67 | 1.16 (0.58–2.28) |
| Chronic kidney disease | 0.64 | 0.78(0.27–2.26) | 0.19 | 0.39 (0.09–1.58) |
| Living with spouse | 0.70 | 1.15 (0.56–2.40) | 0.27 | 0.66 (0.31–1.40) |
| Education of > 6 years | 0.35 | 1.40 (0.70–2.80) | 0.005 | 0.34(0.16–0.71)[#] |
| Smoking | 0.06 | 2.1(0.97–5.48) | 0.41 | 1.50 (0.56–3.97) |
| Alcohol | 0.41 | 0.68(0.27–1.70) | 0.89 | 0.92 (0.33–2.54) |
| Benut | 0.03 | 3.32 (1.12–9.83)[#] | 0.11 | 2.53 (0.82–7.77) |
| Mood disturbance | 0.005 | 5.43 (1.66–17.7)[#] | 0.008 | 3.92 (1.43–10.8)[#] |

OR: Odds ratio, Benut: Areca catechu,

[#]: $p < 0.05$.

with the condition before stroke onset. A higher risk of insomnia symptoms was associated with atrial fibrillation, hypertension, and mood disturbance in the acute stroke period, whereas low education and mood disturbance were linked to insomnia in the chronic stroke periods.

According to the DSM-5 criteria, a patient is diagnosed with insomnia based on the following symptoms: difficulties in initiating sleep, challenges in maintaining sleep, early-morning awaking with inability to return to sleep, and sleeping difficulties lasting at least 3 months. Most studies investigating insomnia were performed at 3 months after stroke onset, and studies prospectively assessing sleep disorders during the acute and chronic stroke period are limited. This was a prospective observational study, and patients reported their sleeping conditions before stroke onset and during the acute and chronic stroke periods. Sleep disturbance during the acute stroke period did not fulfill all the DSM-5 criteria for insomnia because the condition did not last at least 3 months and was considered as insomnia symptom. However, all participants met the following criteria: difficulties in initiating sleep, challenges in maintaining sleep or early-morning awakening with inability to return to sleep, and sleeping difficulties lasting at least 3 nights per week. In this study, patients reported their prestroke sleeping condition within the first week after stroke onset, which is more reliable than other studies in which patients reported such information 3 and 6 months after stroke onset [8, 31].

Moreover, in our study, 34.3% of the patients presented with insomnia before stroke onset, which is lower than that of a population study in Taiwan (41.4%) [32]. In the Tsou study, only patients aged > 65 years were included, which might have caused the differences in results. In the current research, 43.6% of patients were aged < 65 years old and those with a depression history were excluded.

Additionally, during the acute stroke period, 37.4% of the patients presented with insomnia symptoms. The prevalence of insomnia symptoms was slightly higher than that in the study by Glozier et al., which showed that 33.4% of the patients had insomnia within the first month after stroke onset. This difference could be attributed to the fact that the patients in the previous study reported their sleep information later than those in our study.

Our study showed that 56.7% of the patients with a history of insomnia prior to stroke onset presented with the condition during the chronic stroke period. Also, not all patients who

presented with insomnia before stroke onset developed the condition after stroke onset. Totally, 43.3% of the patients with insomnia before stroke onset recovered from the condition 3 months after stroke onset (Fig 1). Moreover, 52% of the patients who initially presented with insomnia symptoms during the acute stroke period continued to experience the condition during the chronic stroke period. The prevalence of insomnia was similar to that in the study by Glozier et al., i.e., 47% of the patients with insomnia within the first month after stroke onset continued to have it 6 months after stroke onset [31]. Furthermore, our study revealed that 15.6% of the patients who did not have insomnia before stroke onset developed the condition after 3 months; thus, insomnia was correlated with stroke. Furthermore, the prevalence of insomnia was similar to that in the study by Leppävuori et al., i.e., 38.6% of the patients with stroke had insomnia before stroke onset, and 56.7% complained about the condition 3–4 months after stroke onset. In 18.1% of the patients, insomnia was considered a sequela of stroke [8]. Approximately half of the patients with insomnia before the onset of stroke experienced the condition 3 months after the onset of stroke. These findings underscore the importance of obtaining information on sleep quality prior to stroke. To improve the quality of life of patients with stroke, it is important to obtain information on sleep quality before stroke and provide individualized treatment.

In total, 29.7% of the patients experienced insomnia 3 months after stroke onset. The prevalence of insomnia among patients with stroke was similar to that in the study by Glozier et al., i.e., 28.3% of the patients had insomnia 6 months after stroke onset.

Based on our study, it could be stated that atrial fibrillation, hypertension, and mood disturbance were associated with a higher risk of sleep disorders in the acute stroke period. Low education and mood disturbance at 3 months after stroke onset were considered a risk factors for insomnia. Moreover, approximately 40% patients with insomnia symptoms in the acute stroke period did not present with the condition after 3 months. Medications, including benzodiazepine and antidepressants were not provided to patients with insomnia in the acute stroke period. The exceptions were those who used hypnotic agents before stroke onset and those with insomnia affecting physical therapy. The condition could be related to anxiety, stress, and treatment in the acute stroke period. Whether insomnia in the acute period but not in the subacute or chronic period is related to reperfusion in the penumbra area needs further investigation.

In most cases, patients with insomnia symptoms in the acute stroke period recovered 3 months after stroke onset. Hence, pharmacotherapy for insomnia in the acute stroke period should be adopted based on individual patient needs [33].

The current study has several strengths. It was a prospective study, and we recorded data on sleeping conditions before stroke onset and during the acute and chronic stroke periods. Moreover, the finding revealed that stroke is not always associated with a higher risk of insomnia and that some patients might recover from the condition. However, this research also has many limitations. Only approximately one third of the patients with stroke were included in the study. We did not include patients with aphasia, previous stroke history, mental impairment, depression, and critical illness, and those who did not agree to enroll in the study, which might have caused selection bias. The diagnosis of sleep disturbance in the acute stroke period did not meet the DSM-5 criteria for insomnia. Moreover, the diagnosis of insomnia was based on the scores of patient self-reported scale and without the use of actinography or polysomnography. We did not evaluate the impact of insomnia on the outcome of patients with stroke. In addition, the number of participants was relatively small. Finally, most patients presented with mild stroke. Nevertheless, those with moderate to severe stroke were also included.

## Conclusion

Insomnia is commonly seen among patients with ischemic stroke. The prevalence rates of sleep disturbance are 34.3%, 37.4%, and 29.7% before stroke onset and during the acute and chronic stroke periods, respectively. The incidence of stroke-related insomnia is 15.6%. Thus, stroke might cause insomnia. However, some patients manage to recover from the condition. A higher risk of sleep disturbance is associated with atrial fibrillation, hypertension, and mood disturbance in the acute and chronic stroke periods. Low education and mood disturbance are associated with the increased risk of insomnia in the chronic stage.

## Acknowledgments

The authors would like to thank the stroke teams in the hospital, patients with stroke, and their relatives for their contribution to the study. We thank Enago (www.enago.tw) for their contribution to English editing.

## Author Contributions

**Conceptualization:** Yi-Sin Wong, Cheung-Ter Ong.

**Data curation:** Hui-Ju Tsai.

**Formal analysis:** Hui-Ju Tsai.

**Investigation:** Hui-Ju Tsai, Yi-Sin Wong.

**Methodology:** Cheung-Ter Ong.

**Project administration:** Hui-Ju Tsai.

**Software:** Yi-Sin Wong.

**Supervision:** Cheung-Ter Ong.

**Visualization:** Yi-Sin Wong.

**Writing – original draft:** Yi-Sin Wong.

**Writing – review & editing:** Cheung-Ter Ong.

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
