## [Decision Letter · Decision Letter 0]

21 Sep 2022

PONE-D-22-17272Clinical course and risk factors for sleep disturbance in patients with ischemic strokePLOS ONE

Dear Dr. Ong,

Thank you for submitting your manuscript to PLOS ONE. After careful consideration, we feel that it has merit but does not fully meet PLOS ONE’s publication criteria as it currently stands. Therefore, we invite you to submit a revised version of the manuscript that addresses the points raised during the review process.

Please revise the manuscript according to the Reviewer's suggestions.

We look forward to receiving your revised manuscript.

Kind regards,

Claudio Liguori

Academic Editor

PLOS ONE

Journal Requirements:

 “This research was supported by Ditmanson Medical Foundation Chia-Yi Christian Hospital Research grant (R109-14).”

“This research was supported by Ditmanson Medical Foundation Chia-Yi Christian Hospital Research grant (R109-14).”

 “This research was supported by Ditmanson Medical Foundation Chia-Yi Christian Hospital Research grant (R109-14).”

Additional Editor Comments:

I suggest the Authors to revise the manuscript according to the Reviewer's suggestions.

Reviewers' comments:

Reviewer's Responses to Questions

**Comments to the Author**

1. Is the manuscript technically sound, and do the data support the conclusions?

Reviewer #1: Partly

2. Has the statistical analysis been performed appropriately and rigorously? 

Reviewer #1: Yes

3. Have the authors made all data underlying the findings in their manuscript fully available?

Reviewer #1: Yes

4. Is the manuscript presented in an intelligible fashion and written in standard English?

Reviewer #1: Yes

5. Review Comments to the Author

Reviewer #1: Authors assessed the risk of insomnia among patients with ischemic stroke and found different patterns of change in sleep habits, including new onset of insomnia and resolution of previous insomnia.

Some points of the methods and the presentation of results should be clarified.

1 - There are some previous studies on the prevalence of insomnia after stroke. I suggest elaborating on the similarities and differences in methods between the present study and previous ones.

2 - Some background information should be better put in context. For example, the general discussion on the core and penumbra seems off topic. It would be better replaced by a discussion on the stroke locations that are more likely to lead to insomnia.

3 - Some factors such as the stress of hospital admission and the acute treatment of stroke may have an impact on sleep habits. This point should be further elaborated.

4 - Authors relied on subjective patient-reported outcomes, in the absence of objective measurements such as actigraphy or polysomnography. This could be a limitation to discuss.

5 - How did Authors deal with patients with cognitive or language dysfunction? Patients must have responded to questionnaires for this study. If Authors excluded from the study patients with language and/or cognitive dysfunction, this would have led to selection bias.

6 - Authors state that most cases of insomnia occurring during the acute phase of stroke tend to resolve within 3 months. For that reason, in the Authors' opinion, it is not recommended to provide sleep medication to patients with stroke in the acute phase. In my opinion, this is not a strong recommendation as some patients with stroke might need sleep medication in the acute phase even if sleep disturbances do not last long. I suggest articulating this point in a better fashion.

7 - I see from the Figure that most patients without insomnia who developed insomnia during the acute phase of stroke resolve their condition, while most patients with previous history of insomnia are likely to remain in their condition after stroke. This point should be better discussed as it has clinical relevance. It would mean that previous history of insomnia should be seeked in patients with stroke to plan adequate interventions in the long term.

8 - Did insomnia influence stroke outcomes in terms of survival and post-stroke disability? Reporting data on this aspect could identify an area of intervention that would lead to benefits in post-stroke recovery.

6. PLOS authors have the option to publish the peer review history of their article (what does this mean?). If published, this will include your full peer review and any attached files.

Reviewer #1: No

---

## [Author Response · Author response to Decision Letter 0]

26 Sep 2022

1 - There are some previous studies on the prevalence of insomnia after stroke. I suggest elaborating on the similarities and differences in methods between the present study and previous ones.

Response:

In introduction, we have added the sentence. “Most studies investigated prevalence of insomnia in patients with ischemic assessing sleep quality after stroke onset. Prospective data on sleep quality in ischemic stroke patients before stroke, acute stroke and subacute stroke are not available.” (Page 5 to page 6).

2 - Some background information should be better put in context. For example, the general discussion on the core and penumbra seems off topic. It would be better replaced by a discussion on the stroke locations that are more likely to lead to insomnia.

 Response:

In introduction, we add the sentence “During the acute stroke period, two dysfunctional areas affect neurological function which may affect sleep quality in patients with ischemic stroke.” (Page 4).

“Whether the neurological impairment in acute stroke phase of stroke may increase risk of insomnia and whether insomnia in acute stroke period many improves after reperfusion in penumbra requires further investigation.” (Page 5).

3 - Some factors such as the stress of hospital admission and the acute treatment of stroke may have an impact on sleep habits. This point should be further elaborated.

 Response:

 In discussion, we add the sentence “The condition could be related to anxiety, stress and treatment in acute stroke period. Whether insomnia in acute period but not in subacute or chronic stroke period related to reperfusion in penumbra area need further investigation.” (Page 14).

4 - Authors relied on subjective patient-reported outcomes, in the absence of objective measurements such as actigraphy or polysomnography. This could be a limitation to discuss.

 Response

In discussion, we have add the limitation

 “The diagnosis of insomnia was based on the scores of patient self-reported scale and absence the measurement of actinography or polysomnography.” (Page 15).

5 - How did Authors deal with patients with cognitive or language dysfunction? Patients must have responded to questionnaires for this study. If Authors excluded from the study patients with language and/or cognitive dysfunction, this would have led to selection bias.

 Response:

 This is the limitation of the study, due to the study depend patient self-report their sleep quality, we have state the limitation in discussion section.

 “we did not include patients with aphasia, a previous stroke history, mental impairment, depression, critical medical disease, and patients who did not agree to enroll in the study, which may have selection bias. However, we have do our best to include the patients who can correctly report their sleep condition.” (Page 15).

6 - Authors state that most cases of insomnia occurring during the acute phase of stroke tend to resolve within 3 months. For that reason, in the Authors' opinion, it is not recommended to provide sleep medication to patients with stroke in the acute phase. In my opinion, this is not a strong recommendation as some patients with stroke might need sleep medication in the acute phase even if sleep disturbances do not last long. I suggest articulating this point in a better fashion.

 Response:

 We have changed to “In most cases, patients with insomnia symptoms in the acute stroke period recovered 3 months after stroke onset. Hence, pharmacotherapy for insomnia in the acute stroke period should be adopted individually34.” (Page 14).

7 - I see from the Figure that most patients without insomnia who developed insomnia during the acute phase of stroke resolve their condition, while most patients with previous history of insomnia are likely to remain in their condition after stroke. This point should be better discussed as it has clinical relevance. It would mean that previous history of insomnia should be seeked in patients with stroke to plan adequate interventions in the long term.

 Response:

 In discussion, we add the sentences

 “Approximately half of patients with insomnia before stroke onset experienced insomnia at 3 month after stroke onset. The result underscore the importance of obtaining information on sleep quality prior to stroke. For improving the quality of life of stroke patients, it is Important to obtain the information of sleep quality before stroke and carry individualized treatment.” (Page 13 to page 14). 

8 - Did insomnia influence stroke outcomes in terms of survival and post-stroke disability? Reporting data on this aspect could identify an area of intervention that would lead to benefits in post-stroke recovery.

 Response:

 Because of the study was designed to investigate the prevalence of insomnia in stroke patients, we did not record the outcome of stroke patients. At present, we cannot report the relationship between insomnia and outcome of stroke. This will be the subject of our next research.

---

## [Decision Letter · Decision Letter 1]

4 Oct 2022

PONE-D-22-17272R1Clinical course and risk factors for sleep disturbance in patients with ischemic strokePLOS ONE

Dear Dr. Ong,

Thank you for submitting your manuscript to PLOS ONE. After careful consideration, we feel that it has merit but does not fully meet PLOS ONE’s publication criteria as it currently stands. Therefore, we invite you to submit a revised version of the manuscript that addresses the points raised during the review process. Please revise the English as requested by the Reviewer and respond to the minor comments raised.

We look forward to receiving your revised manuscript.

Kind regards,

Claudio Liguori

Academic Editor

PLOS ONE

Journal Requirements:

Reviewers' comments:

Reviewer's Responses to Questions

**Comments to the Author**

1. If the authors have adequately addressed your comments raised in a previous round of review and you feel that this manuscript is now acceptable for publication, you may indicate that here to bypass the “Comments to the Author” section, enter your conflict of interest statement in the “Confidential to Editor” section, and submit your "Accept" recommendation.

Reviewer #1: All comments have been addressed

2. Is the manuscript technically sound, and do the data support the conclusions?

Reviewer #1: Yes

3. Has the statistical analysis been performed appropriately and rigorously? 

Reviewer #1: Yes

4. Have the authors made all data underlying the findings in their manuscript fully available?

Reviewer #1: No

5. Is the manuscript presented in an intelligible fashion and written in standard English?

Reviewer #1: No

6. Review Comments to the Author

Reviewer #1: Authors duly addressed the comments raised. Some minor points remain for improvement.

1 - I understand that the "chronic stage" of stroke is 3 months. In my opinion, it would be better to replace "chronic stage" with a more precise time indication.

2 - The lack of outcome data on stroke should be stated as a major limitation of the study.

3 - English style should be carefully revised as there are many grammatical imprecisions.

7. PLOS authors have the option to publish the peer review history of their article (what does this mean?). If published, this will include your full peer review and any attached files.

Reviewer #1: No

---

## [Author Response · Author response to Decision Letter 1]

14 Oct 2022

We thank reviewer for their detail review and very important comments. Base on reviewer’s comments, we have made necessary changed point by point.

1 - I understand that the "chronic stage" of stroke is 3 months. In my opinion, it would be better to replace "chronic stage" with a more precise time indication.

Response: For precise timing indication, we have used chronic stage replace subacute stroke stage.

2 - The lack of outcome data on stroke should be stated as a major limitation of the study.

Response: We have add the limitation in discussion. In page 15, limitation section, we add the sentence “We did not evaluate the impact of insomnia on the outcome of patients with stroke.” 

3 - English style should be carefully revised as there are many grammatical imprecisions.

Response: The English style of manuscript have been edited again.

---

## [Editor Report · Decision Letter 2]

24 Oct 2022

Clinical course and risk factors for sleep disturbance in patients with ischemic stroke

PONE-D-22-17272R2

Dear Dr. Ong,

We’re pleased to inform you that your manuscript has been judged scientifically suitable for publication and will be formally accepted for publication once it meets all outstanding technical requirements.

Kind regards,

Claudio Liguori

Academic Editor

PLOS ONE

Additional Editor Comments (optional):

The Authors provided the requested revision.